# Temporal and Spatial Distribution of Microplastics in a Coastal Region of the Pearl River Estuary, China

**Siyang Li** [1,2], **Yilin Wang** [1], **Lihong Liu** [1], **Houwei Lai** [1], **Xiancan Zeng** [1,2], **Jianyu Chen** [1], **Chang Liu** [1,*] **and Qijin Luo** [1,*]

[1] South China Institute of Environmental Science, Ministry of Ecology and Environment, Guangzhou 510345, China; lisiyang@scies.org (S.L.); wangyilin@scies.org (Y.W.); liulihong@scies.org (L.L.); laihouwei@scies.org (H.L.); zengxiancan@scies.org (X.Z.); chenjianyu@scies.org (J.C.)

[2] Guangzhou Huake Environmental Protection Engineering CO., Ltd., Guangzhou 510655, China

\* Correspondence: liuchang@scies.org (C.L.); luoqijin@scies.org (Q.L.); Tel.: +86-020-85620700 (C.L.); +86-020-29119680 (Q.L.)

**Abstract:** This study conducted an analysis of microplastics (MPs) pollution in a coastal region of the Pearl River Estuary (PRE) in the South China Sea. The results show that the abundance of MPs during the rainy season reached 545.5 particles m$^{-3}$, which was 1.85-fold higher than during the dry season. The spatial distribution of MPs also varied offshore in the following order: the river > estuary > sea. The average abundance of MPs in the river was 1.17-fold higher than that of the estuary and 4.65-fold higher than that of the marine environment. There were large amounts of gray, white, and green MPs, and about 53.5–73.9% of the MPs were less than 0.5 mm. The main forms of MPs were fibers, granules, fragments, and films. MPs composed of polyethylene accounted for 35.7–38.8%. PCA analysis showed that MPs carried by the river were an important source of MP pollution in the coastal waters.

**Keywords:** microplastics; Pearl River Estuary; coastal pollution; marine environment

## 1. Introduction

Plastics, derived from petroleum, are a synthetic polymer with relatively high molecular weight and plasticity. The global production of plastics has continuously increased since their production began on an industrial scale in the 1950s, and is driven by the use of plastic products for a wide range of purposes [1]. Global plastic production reached 359 million tons in 2018 [2]. These plastics will be degraded and broken down in the natural environment by physical degradation processes, photo- and thermo-oxidative processes and, to a lesser extent, biodegradation, which weakens material integrity [3]. Generally, researchers refer to plastic particles smaller than 5 mm in maximum diameter as microplastics (MPs) [4]. MPs have been found in global aquatic environments (e.g., oceans, estuaries, rivers, lakes, reservoirs) [5,6], the atmosphere [7], and soils [8]. They have even been found in inaccessible regions of Antarctica [9], the Arctic Ocean [10] and even in high-mountain lakes [11], which demonstrates the transport of MPs over long distances through the forces of wind, rivers, and ocean currents.

MPs in the natural environment are often classified according to their source as primary MPs and secondary MPs [12]. Primary MPs are often identified as microscopic plastic particles that are composed of the raw materials for manufacturing plastic products, and microbeads used in human health and personal care products [13]. It is estimated that approximately 306.8 t of MP beads have entered the environment due to the use of personal care products [14]. Secondary MPs are usually derived from weathering and degradation of plastic products, most of which have not been properly handled or discarded. For example, the large-scale use of disposable plastic packaging materials (e.g., plastic bags, plastic tape) in the shipping industry, when unsupervised, produces large amounts of

plastic waste [15]. Synthetic textiles (such as clothing) also produce secondary MPs due to wear and tear during daily use and cleaning [16,17]. The friction between automobile tires and the ground during driving produces large amounts of secondary MPs that enter the environment [18]. Paul et al. [19] found that on average wastewater treatment plants (WWTPs) removed 88% of MPs when applying preliminary/primary plus secondary treatment, and they removed 94% of MPs when applying preliminary, primary, secondary, and tertiary treatments. Although overall removal is high, the residual amount in the treated effluent ($\sim$10% of the MPs in the influent wastewater) represents an important release of MPs to the aquatic environment. Globally, a total of $3.4 \times 10^4$ to $2.0 \times 10^{10}$ MPs particles are released per day [19].

Within this context, the main objective of this study is to investigate the occurrence and distribution of MPs pollution in aquacultural water in the Pearl River Basin near the South China Sea. More specifically, this study provides insights into the occurrence and characteristics of MPs in aquacultural water in this coastal region of the Pearl River Estuary.

## 2. Materials and Methods

### 2.1. Study Area

The Pear River Estuary is a subtropical estuary in South China, along the northern boundary of the South China Sea. This area is the most economically developed region in China; GDP reached USD $1.34 \times 10^3$ billion in 2019. This estuary receives waters from the Pearl River Basin, the third largest basin in China. It is estimated that the annual amount of MPs flowing into the South China Sea from the Pearl River could reach $2.31 \times 10^4$ t yr$^{-1}$ [20].

### 2.2. Collection of Water Surface Samples

Sampling was performed in September (rainy season) and November (dry season) in 2020. Samples were collected from 14 survey sites from the water surface (0–2 m) using a type III plankton net following the method by Kovač Viršek et al. [21]. The net measured 30 cm in width and 20 cm in height at mouth opening, whereas the bag was 100 cm long. The net had a mesh size of 300 μm). The location of the sites are shown in Figure 1 and Table A1. A mechanical flow meter (438 110, HYDRO-BIOS(Erkelenz), Germany) was used to record the amount of filtered water that passed through the net. After the 10–20 min collection time, the valve at the bottom of the net was opened and sea water was used to rinse the net from the outside to the inside repeatedly until all the MP particles were collected in a glass sample bottle. One sample was collected at each sampling point. The samples were then placed in a cooler and transported to the laboratory where they were kept cool and in the absence of light.

### 2.3. Sample Processing and Analysis

In the laboratory, sample processing followed the local methods used in Liaoning Province (DB21/T 2751–2017) China, where MPs were determined in seawater using micro-Fourier Transform Infrared Spectroscopy. Briefly, the water samples to be tested were thoroughly shaken to homogenize the distribution of MPs within the sample. A vacuum filtration device was subsequently used to filter the water sample (100 mL) through a microporous polycarbonate filter membrane; the negative pressure during the filtration process did not exceed 40 KPa to prevent the filter membrane from breaking. Subsequently, the residue on the filter membrane was rinsed and resuspended in 5 mL of deionized water, after which 20 mL of 0.05 mol L$^{-1}$ ferrous sulfate solution and 20 mL of a 30% hydrogen peroxide solution was added to the sample. After an appropriate amount of nitric acid was added to adjust the solution pH to 5–6, the sample was placed on a magnetic stirrer and kept cool at 25–30 °C. Following the generation of bubbles, the beaker was removed and allowed to cool; this operation was repeated until the sample was clear. Finally, a vacuum filtration device was used to filter the sample onto a microporous polycarbonate filter

membrane, and toothless stainless-steel tweezers were used to remove the filter membrane, which was dried and put in the sample box for later use.

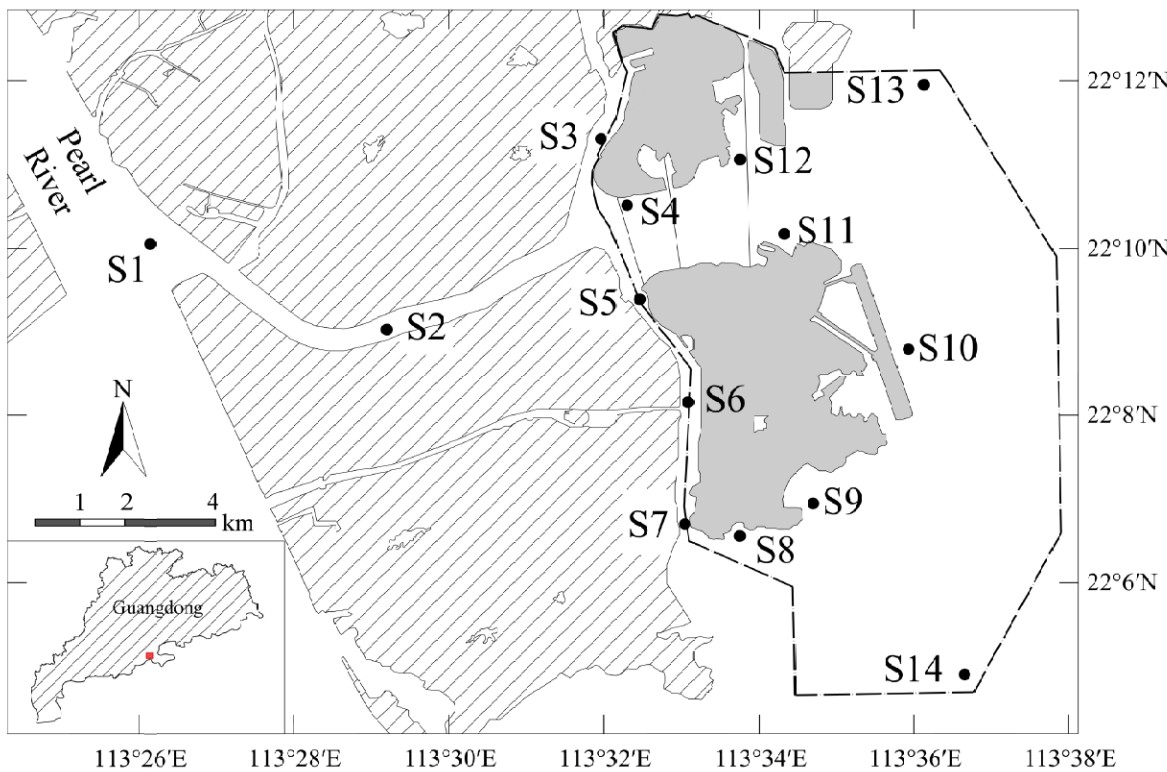

**Figure 1.** Map of the study area showing location of sampling sites.

*2.4. MPs Identification*

Each filter membrane enriched with MPs was examined under a stereomicroscope (SMZ1270, Nikon (Tokyo), Japan; 16×) to locate and identify the MP particles, and a digital camera was used to image the MPs phase. Subsequently, a micro-Fourier Transform Infrared spectrometer (micro-FTIR) equipped with attenuated total reflectance (ATR) (Spectrum Spotlight 400, PerkinElmer Instruments (Waltham, MA, USA) was used to count and measure the size of the polymers invisible to the naked eye. Finally, the data were compared with the Fourier Infrared Spectroscopy database to determine the type and composition of the MPs [22,23].

An infrared microscopy imaging system was used to observe the MP particles on the surface of the filter membrane, and the number of MP particles observed in the field of view ($N_2$) was recorded. The number of MP particles ($N$) was calculated as:

$$N = N_1 + S_0/S \times N_2 \tag{1}$$

where $N$ is the number of MP particles (no. of particles); $N_1$ is the number of MP targets visible to the naked eye (no. of particles); $N_2$ is the number of MP particles detected in the field of view of the microscopic infrared imaging system ($S_0$) (no. of particles); $S$ is the area of the filter membrane (mm$^2$); $S_0$ is the area observed by the microscopic infrared imaging system (mm$^2$). The abundance of MP particles in the water sample was calculated as:

$$A = N/V \tag{2}$$

where $A$ is the abundance of MPs in water (particles m$^{-3}$); $N$ is the number of MP targets (no. of particles); $V$ is the total volume of filtered water (m$^3$).

*2.5. Data Analysis*

The abundance of MPs in the surface water environment of each sampling site was expressed by the number of particles per $m^3$ (particles $m^{-3}$). Microsoft Office Excel 2019, SPSS 19.0 statistical package software (IBM, Armonk, NY, USA) and Origin 9.0 were used to process the data and plot charts. ArcGIS 12.0 was used to plot the distribution of sampling sites. Principal component analysis (PCA) was applied to estimate the sources and contributions of the MPs for rain and dry season. PCA analysis was analyzed using the Canoco 5.04 software.

## 3. Result and Discussion

*3.1. Abundance*

Quantitative analysis of the concentration of collected MPs in the coastal environment of the Pearl River estuary was determined (Figure 2) and compared between different times. The abundance of MPs during the dry season was lower than that in the rainy season, but the absolute value of the abundance was high during both periods. In the rainy season, the abundance of MPs varied from 87.4 to 1790.5 particles $m^{-3}$, with an average value of 545.4 particles $m^{-3}$. Among the sampling sites, S14 had the lowest abundance of 87.4 particles $m^{-3}$, and S3 had the highest abundance of 1790.5 particles $m^{-3}$. The abundance of MPs in the dry season ranged from 69.7 to 803.6 particles $m^{-3}$, with an average value of 294.6 particles $m^{-3}$. Site S14 had the lowest abundance of 69.7 particles $m^{-3}$, whereas S1 had the highest abundance of MPs, with a value of 803.6 particles $m^{-3}$.

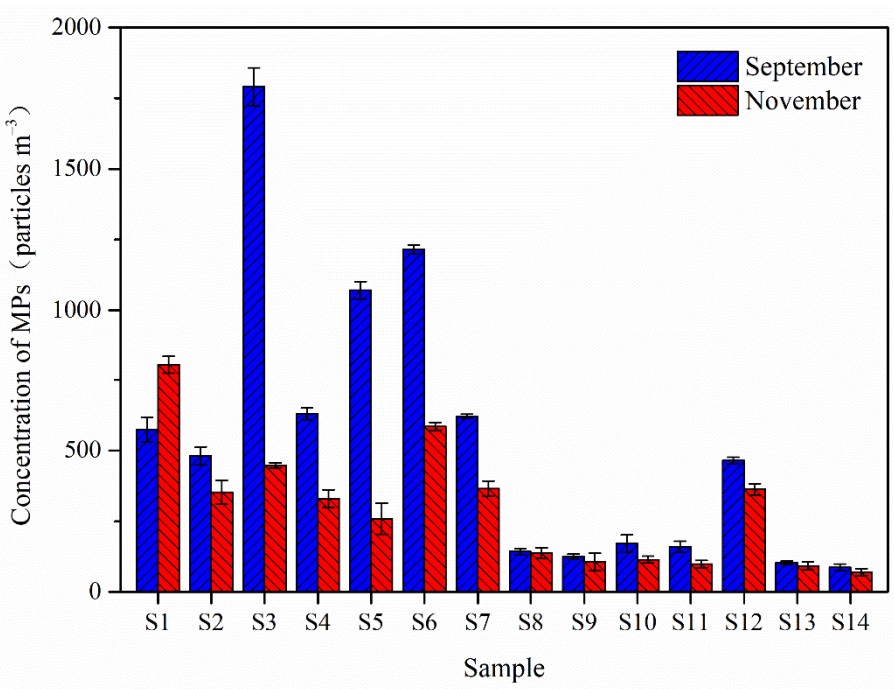

**Figure 2.** Concentrations of MPs in a coastal region of the PRE in rain and dry season.

The spatial distribution of MP pollution in the coastal waters was distinct (Table 1). The abundances of MPs significantly decreased from the river (S1–S3), to the estuary (S4–S7), and to the sea (S8–S14); the average values in these three environments were 741.5 particles $m^{-3}$, 634.6 particles $m^{-3}$, and 159.6 particles $m^{-3}$, respectively. The abundance of MPs in riverine environments was 1.1–1.4-fold that of the estuary and 4.8–5.3-fold that of sampled areas of the sea during rainy and dry seasons. Only S1 (river) shows higher MPs during the season, this phenomenon proves that the source of the MPs is the river, and that its stream to the sea during the dry season is slower. Moreover, the above results demonstrate that MPs carried by the upstream river water to the estuary was an important source of MP pollution. At the same time, we observed that the abundances of MPs at site

S12 during both the rainy and dry seasons were significantly higher than those located further offshore in area of sea water (S8–S14), reaching 362.4–464.9 particles m$^{-3}$. This may be due to the discharge of MPs from the outlet of a sewage treatment plant near S12, which was similar to the results of Wang et al. [24].

**Table 1.** The spatial distribution of microplastics in a coastal region of the PRE.

| Area * | Rain Season | | Dry Season | |
|---|---|---|---|---|
| | **Range** | **Average** | **Range** | **Average** |
| River | 481.2–1790.5 | 948.5 | 352.6–803.6 | 534.5 |
| Estuary | 629.6–1215 | 884.3 | 259.2–585.0 | 385.0 |
| Sea | 87.4–264.9 | 150.5 | 69.7–162.4 | 111.5 |

* River: S1–S3; Estuary: S4–S7; Sea: S8–S14.

Because there is no universal standard for evaluating MPs pollution, and factors such as sampling method, filter-membrane pore size, and MP particle size varied between studies, it is difficult to directly compare the MPs abundance to previous investigations. However, some results obtained from studies using similar research methods were selected for discussion (Table 2). The abundance of MPs varies greatly between regions (Table 2) due to differences in human activities as well as hydrological and hydrodynamic conditions. The abundance of MPs in the coastal waters of this study area was higher than that in the waters of southern Sri Lanka [25], and similar to the coastal waters of South Korea [26]. Regional economic development and human activities have a strong influence on MP pollution [27]. However, the abundance of MPs in the coastal waters of the study area was significantly lower than that of coastal areas in the UK [28]. The difference is likely due to the particle size considered. In the latter study, the range of MPs analyzed was 0.005–5 mm, which allowed the relatively small MP particles in the sample to be collected, leading to higher MP concentrations. Compared with the results of other freshwater rivers and estuaries in China, the abundance of MPs in the coastal waters of the studied region was lower [29,30]. These differences may be due to the effects of higher levels of human activity on inland (freshwater) water bodies, and to the hydrological and hydrodynamic conditions in freshwater systems, which are not as conducive to diffusion and dilution. Compared with the findings in Hong Kong, another special administrative region in the Pearl River Estuary of China, the MP pollution documented in this study was more serious [31]. These data may indicate that in addition to human activities, hydrological and hydrodynamic conditions are also an important factor affecting regional MP contamination. Compared with MP pollution in estuary and lake sediments, the abundance of MPs in the aquatic environment was lower [32,33], suggesting that although MPs may be transported in the aquatic environment due to their size, shape, and composition [34,35], MPs will gradually settle and eventually enter the sediments [36].

*3.2. Spatial Distribution of MP Characteristics*

Studies have shown that classifying the appearance (e.g., shape, color, and size) and surface morphological characteristics of MPs may help to further trace the source of MP pollution. The general characteristics of MPs in this study are shown in Figure 3. Fibers were the most abundant morphotype, comprising 62.8–81.1% and 40–82.4% of the samples in the rainy and dry seasons, respectively. This finding is consistent with the result of André et al. [5]. Granules were the second most abundant morphotype, accounting for 5.1–21.2% in the rainy season and 6.5–40% in the dry season, respectively. Only a small number of MPs were identified as fragments (2.7–19.8% in the rainy season and 4.7–20.4% in the dry season). Films were rarely found, accounting for only 2.0–7.5% and 0–7.5% in the rainy and dry seasons, respectively (Figure 3a,b).

**Table 2.** Current status of domestic and international MPs pollution in different regions.

| Location | Regions | Sample Type | Microplastic Particle Detection Range (mm) | MP Abundance | Source * |
|---|---|---|---|---|---|
| Sri Lanka | Southern seas | Ocean | 1.5–2.5 | 0–29 | (Koongolla et al., 2018) |
| Korea | Coastal oceans | Ocean | 0.02–5 | 448–2000 | (Song et al., 2018) |
| Britain | Coastal oceans | Ocean | 0.005–5 | 1500–6700 | (Li et al., 2018) |
| China | Wuhan | Lake | 0.05–5 | 1160–8925 | (Wang et al., 2017) |
| | Yangtze River | River | 0.03–5 | 500–10,200 | (Zhao et al., 2014) |
| | Sanjiangkou | Estuary | 0.33–5 | 100–4100 | (Zhao et al., 2015) |
| | Danjiangkou | Reservoir | 0.05–5 | 467–15,017 | (Di et al., 2019) |
| | Hong Kong | Ocean | 0.03–4.96 | 0.51–279.1 | (Tsang et al., 2017) |
| | Boyang Lake | Sediment | 0.002–5 | 11–3153 * | (Liu et al., 2019) |
| | Hong Kong | Sediment | 0.135–5 | 106–15,554 ** | (Fok and Cheung, 2015) |
| India | Vembanad Lake | Sediment | less than 5 | 252.8 ** | (Sruthy and Ramasamy, 2017) |

* particles kg$^{-1}$; ** particles m$^{-2}$.

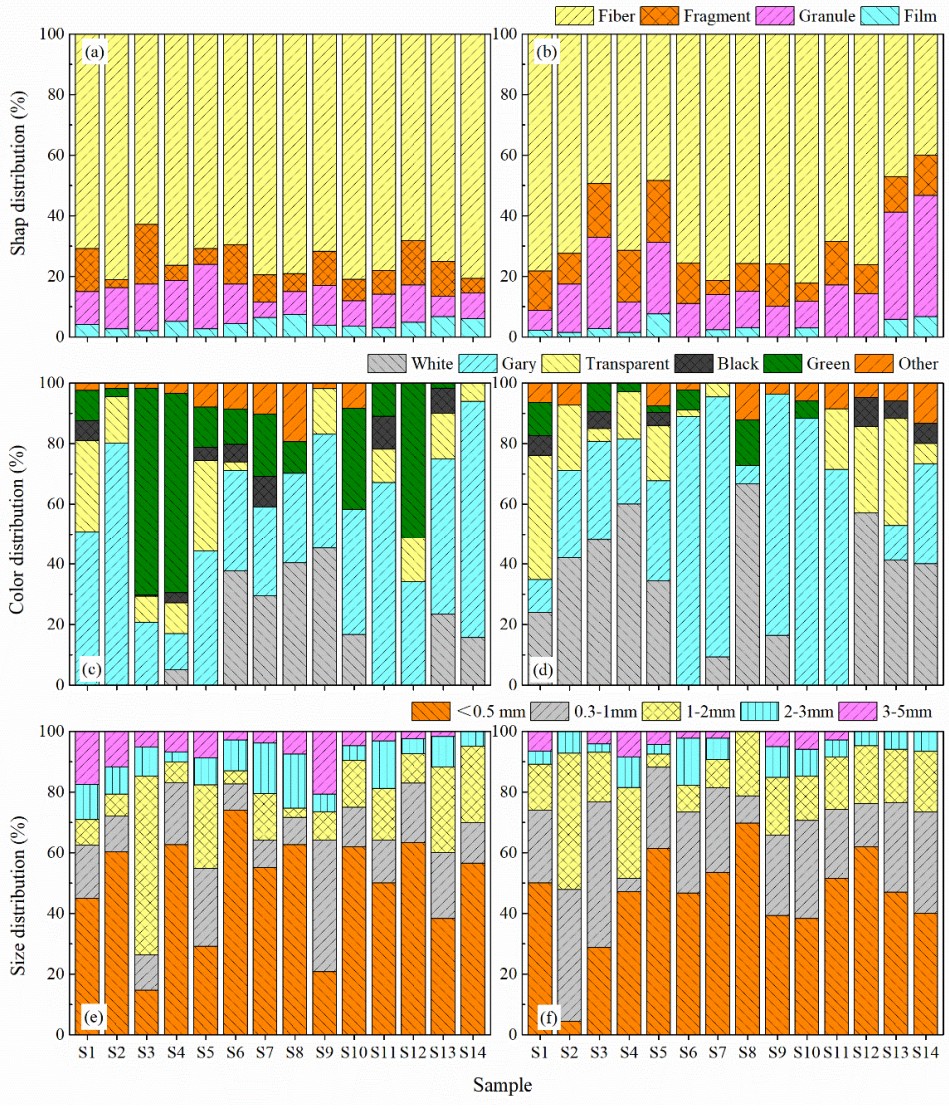

**Figure 3.** Spatial variations in MP shape: (**a**–Rain season, **b**–Dry season), color (**c**–Rain season, **d**–Dry season) and size (**e**–Rain season, **f**–Dry season).

The color of the MP particles can indicate the material and source of the MP particles to some extent. Therefore, MP particles in the coastal waters of this area were classified into five different colors, including white, gray, transparent, black, and green; their temporal and spatial distribution is shown in Figure 3c,d. In the rainy season, gray and green MPs accounted for the largest proportion of the total; the average values at each site reached 43.7% and 21.5%, respectively. In the dry season, gray and white MPs were dominant, accounting for 42.4% and 31.4%, respectively. The occurrence of different colors of MPs detected in this sampling survey may be due to the fact that the Pearl River, which is located adjacent to the study area, is the second largest water system in China. The socio-economic development in the area is also high. Thus, a wide range of plastic waste types may be weathered and broken during transportation along the Pearl River by surface runoff before ultimately entering the sea.

The size of MP particles can determine whether they are easily ingested by biota and enter the food chain. Therefore, the analysis of the size of MPs helps to assess the biological risk of MP pollution [37]. In this study, MPs were classified into five size classes: <0.5 mm; 0.5–1 mm; 1–2 mm; 2–3 mm; 3–5 mm. The temporal and spatial distribution of these MP size classes are shown in Figure 3e,f. During the rainy and dry season sampling campaigns, MP particles smaller than 0.5 mm accounted for 47.3% and 45.4% of the total, respectively. The samples collected at S6, S7, and S12 near the sewage treatment plant in rainy and dry seasons contained significantly higher amounts of MPs smaller than 0.5 mm than those at other sites, accounting for 53.5–73.9% and 61.9–63.4%, respectively (Figure 3e,f). The abundances of MPs near the two sewage treatment plants were also relatively high during the two surveys, with the highest values reaching 1215.1 and 464.9 particles $m^{-3}$ in rainy and dry seasons, respectively. The above findings suggest that the effluent from the sewage treatment plant is an important source of MP pollution, and is consistent with previous studies that have found that the effluent from treatment plants is composed of relatively small amounts of large-sized MPs and high quantities of small-sized MPs [19].

The four shapes of MPs could be defined under the stereomicroscope (Figure 4). Among them, MP fibers exhibit a long and thin shape (Figure 4a,b). These fibers are likely to be derived from fabric (clothing). The edges of these fibers were relatively smooth (Figure 4c,d), and the surface of the fragments generally had signs of weathering and cracking (Figure 4e,f), potentially indicating that the fragments were produced from the breakage of large pieces of plastic products [29] or abrasion of industrial raw materials during transport, particularly in rivers. The films were composed of soft plastic fragments (Figure 4g,h), and they might be generated from the degradation of plastic bags, or plastic wrap, among other products. The surface morphology of MPs was further observed by scanning electron microscopy (SEM) (Figure 5). Fibrous MPs showed a distorted or stretched condition (Figure 5a,b), whereas granular MPs exhibited different surface characteristics; they exhibited rough (Figure 5c), smooth (Figure 5e), and regular surface morphology (Figure 5e,g–i). The appearance of MP fragments exhibited severe erosion characteristics, such as irregular porous structures (Figure 5j,k), and severe mechanical fracture or natural weathering patterns (Figure 5l–n). The surface of the MP films mostly had obvious and regular pore structures (Figure 5o,p).

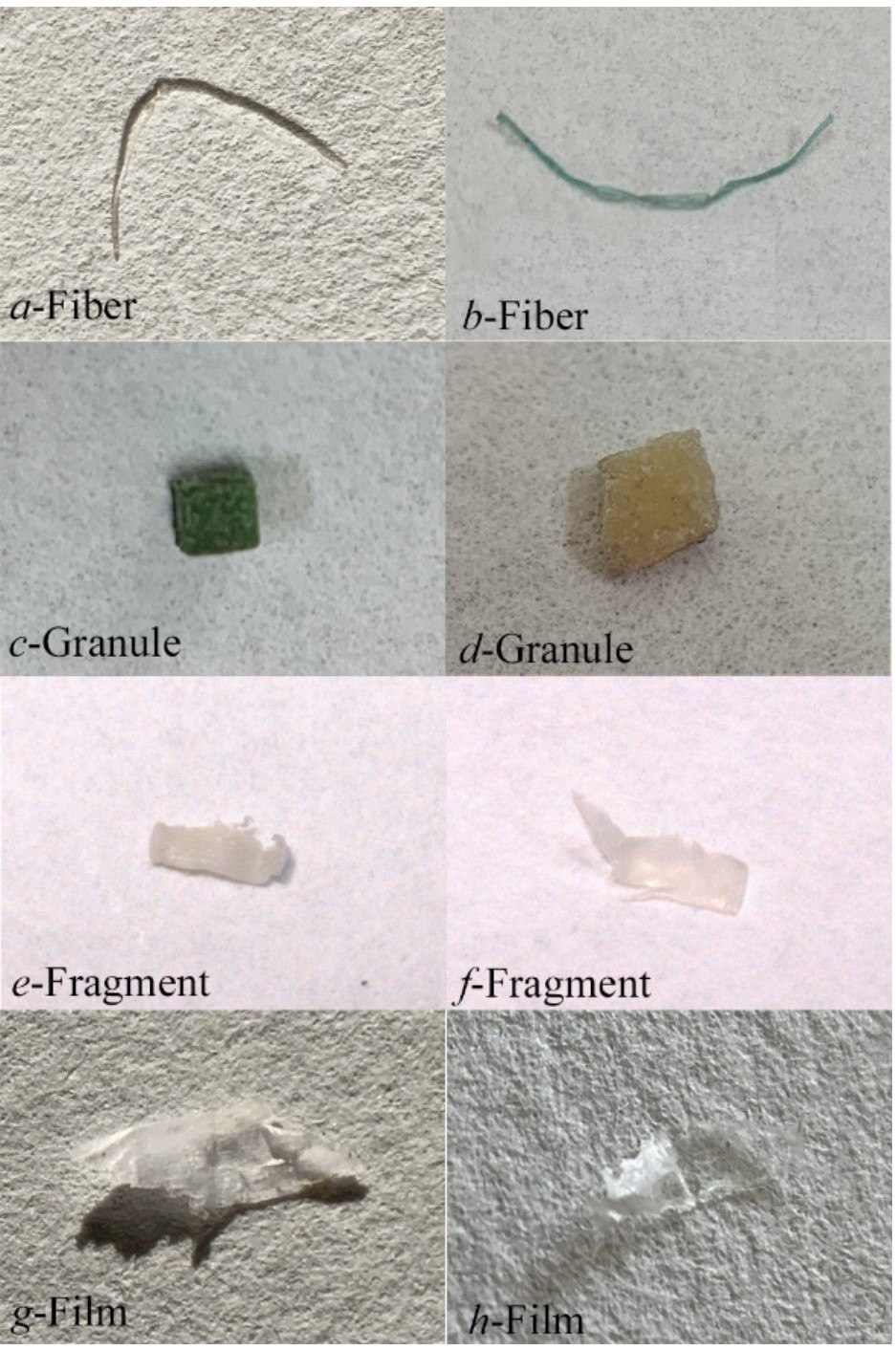

**Figure 4.** Photographs of typical MPs of varying shape collected from surface waters in a coastal region of the PRE: (**a**,**b**–Fiber, **c**,**d**–Granule, **e**,**f**–Fragment, **g**,**h**–Film).

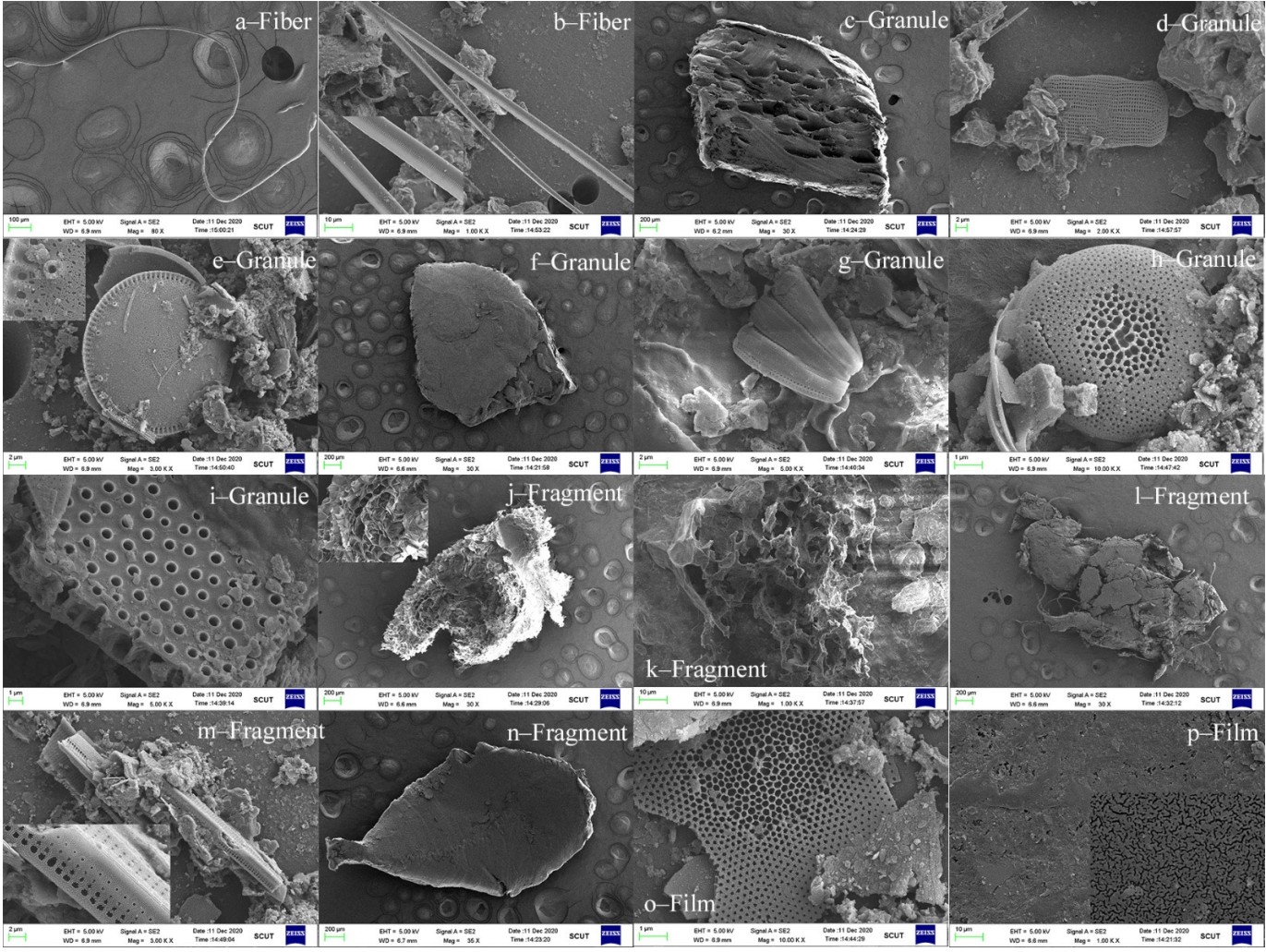

**Figure 5.** SEM photomicrographs of selected MP particles.

Principle component analysis (PCA) (Figure 6) [2], showed that during the rainy season, sites S1, S2, S3, S5, and S11 were located in the first and fourth quadrants. Most of these sites were located in river areas and estuary areas, and were strongly related by the color, size, and appearance of the MPs. The PCA results demonstrate that MP pollution at these sites was more serious than in the sea (further offshore). Similarly, in dry season, sites S1, S2, S3, S4, and S5 belonging to river areas and estuary areas were located in the fourth quadrant. The above results were similar to the results in Section 3.1, suggesting that MPs carried by the river are one of the important sources of MPs in the coastal waters of the region.

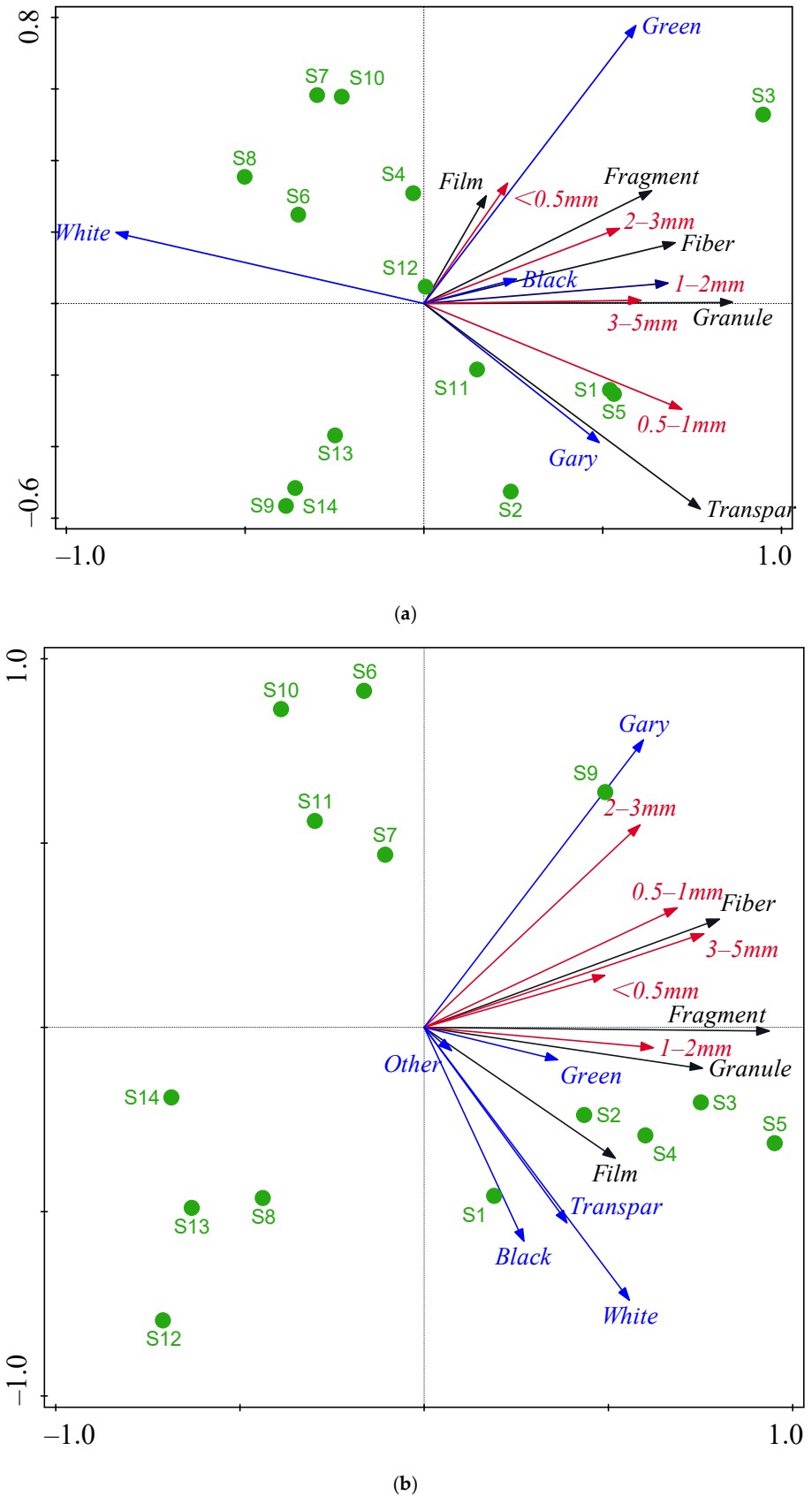

**Figure 6.** Principal component analysis (PCA) plot for MPs: (**a**–Rain season, **b**–Dry season).

### 3.3. MPs Composition

Suspected MP particles were selected and identified by micro–FTIR (Figure 7). Based on the selected samples, MPs identified around the study area of the PRE were primarily composed of polyethylene (PE), polyvinyl chloride (PVC), polypropylene (PP), polystyrene (PS), and ethylene/acrylic acid copolymer (EAA). During both surveys, the monthly average proportions of PE were the highest (38.7% and 35.7% in the rainy and dry seasons, respectively). The highest values were 80% (S1) and 67.1% (S3) (Figure 7). Fibers were primarily comprised of PVC and PP, while granules were comprised of EAA and PS. Fragments were comprised of PS and PVC, while films were comprised of PP and PE, respectively (Figure 8). Polyethylene is one of the most commonly used raw materials in plastic production. It is resistant to wear, exhibits high plasticity and is widely used in the manufacture of toys, bottles, pipes, household plastic utensils, and food packaging bags [38]. Polyvinyl chloride is a general-purpose plastic material that was once the most extensively produced plastic globally. It is widely used as construction materials, industrial products, and household goods [39]. Plastics made of PE and PVC are subject to the weathering and erosion in the natural environment, and form MPs composed of relatively small size particles which can enter the food chain through their ingestion by fish and other aquatic organisms [40]. Thus, attention pertaining to their migration, accumulation, and toxic effects in the food chain are needed [37].

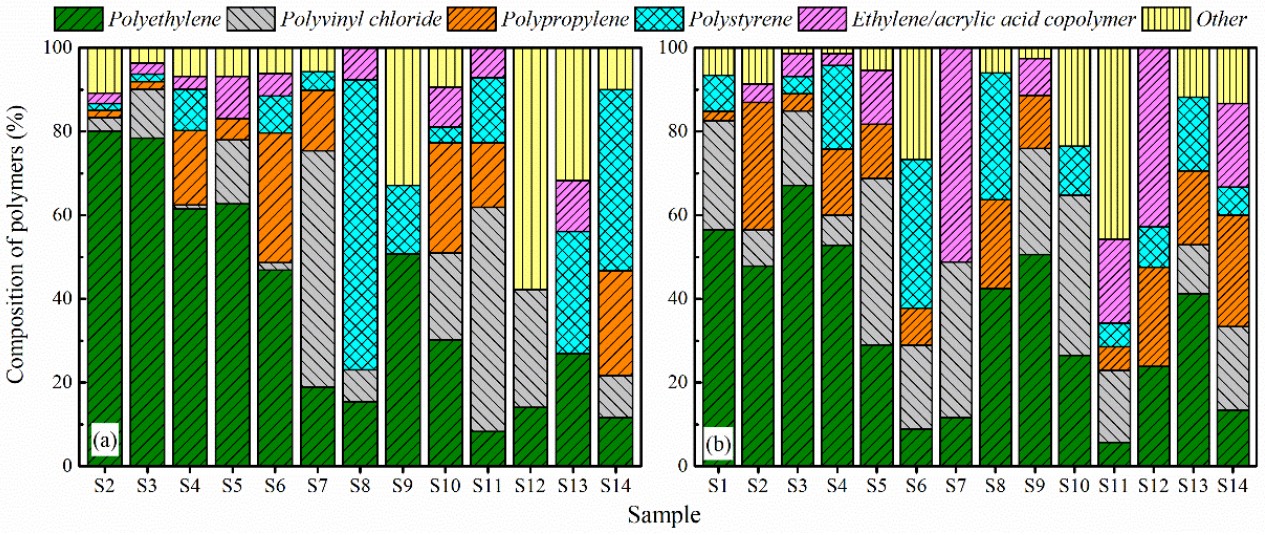

**Figure 7.** Composition of MPs collected in a coastal region of the PRE: (**a**–Rain season, **b**–Dry season).

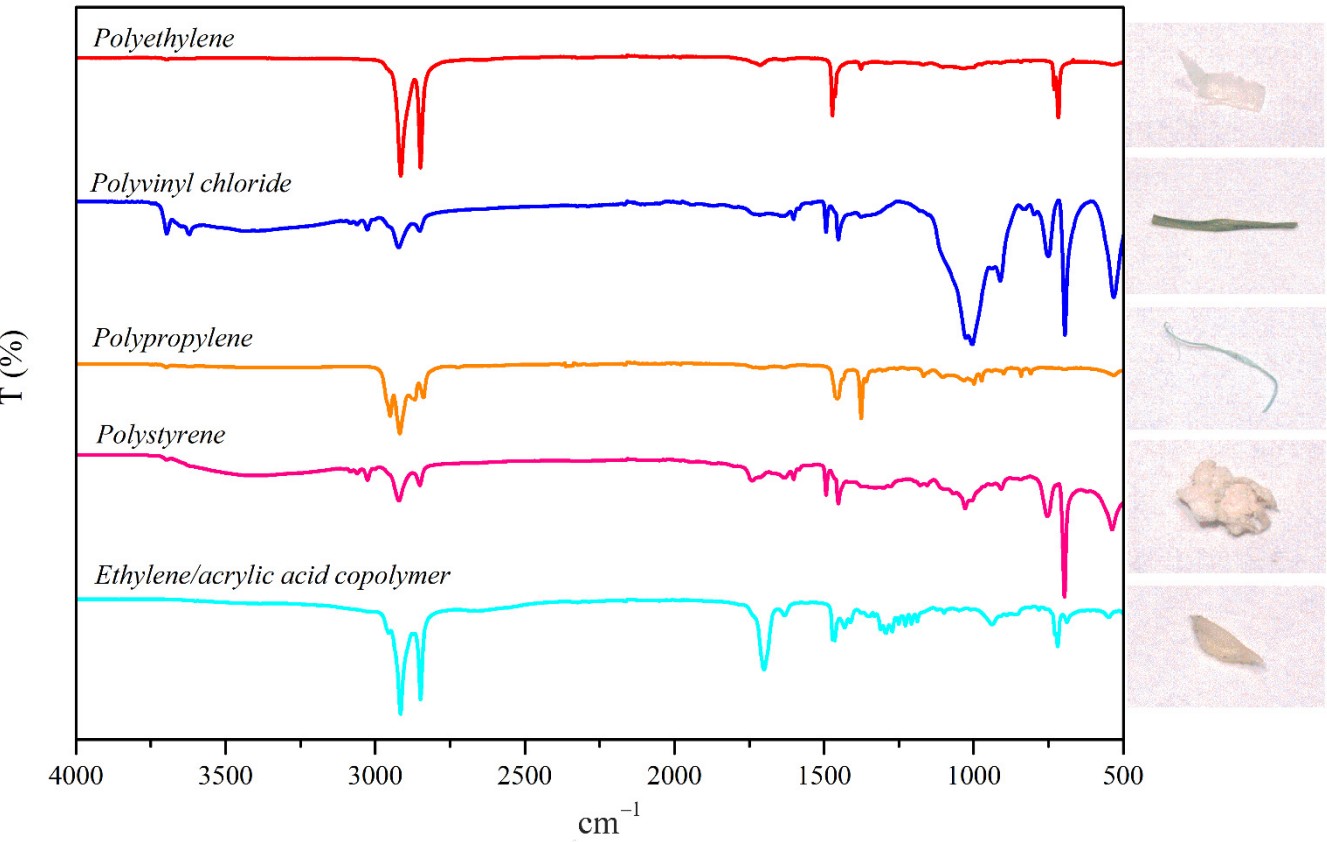

**Figure 8.** MPs compositions of selected samples identified by micro–FTIR.

### 4. Conclusions

This study investigated the current status of MPs pollution in the surface waters of a coastal region in the South China Sea. The results of the survey showed that the region was greatly affected by MPs from the Pearl River and the effluent from a sewage treatment plant. Long-term and large-scale monitoring of MPs in dense urban areas in the middle and lower reaches of the Pearl River Basin should be conducted to more fully understand the transport, source, and fate of MPs within the land–sea ecosystem. MP particles detected in this survey exhibited a range of colors, and nearly half of the particles were less than 0.5 mm in size, indicating that these MPs may be accidentally ingested by fish, shrimp, shellfish, and other organisms and enter the food chain. Most MPs were composed of PE and PVC, which may be carcinogenic. Therefore, they are likely to endanger human health after entering the food chain. As a result, further study on the accumulation and transfer of contaminants associated with MPs between trophic levels, and the biological toxicity of MPs, is needed.

**Author Contributions:** Conceptualization, C.L. and Q.L.; methodology, S.L. and C.L.; software, S.L.; validation, C.L. and Q.L.; formal analysis, S.L.; investigation, Y.W., L.L. and H.L.; resources, J.C.; data curation, X.Z.; writing—original draft preparation, S.L.; writing—review and editing, C.L.; visualization, S.L.; supervision, J.C.; project administration, J.C.; funding acquisition, C.L. and Q.L. All authors have read and agreed to the published version of the manuscript.

**Funding:** This research was funded by the Key-Area Research and Development Program of Guangdong Province, grant number 2020B1111380003 and the Fundamental Research Funds for the Central Public Welfare Research Institutes, grant number PM-zx097-201904-131.

**Institutional Review Board Statement:** Not applicable.

**Informed Consent Statement:** Not applicable.

**Data Availability Statement:** Not applicable.

**Acknowledgments:** We would like to thank Cheng Qian and Jun Li (PerkinElmer, Inc.) for providing MPs testing equipment and technical support.

**Conflicts of Interest:** The authors declare no conflict of interest.

## Appendix A

**Table A1.** The GPS message of sampling point.

| No. | Longitude | Latitude |
| --- | --- | --- |
| S1 | 113°26′37.21″ | 22°10′02.17″ |
| S2 | 113°28′57.19″ | 22°09′23.42″ |
| S3 | 113°32′06.25″ | 22°11′51.77″ |
| S4 | 113°32′25.14″ | 22°10′32.51″ |
| S5 | 113°32′18.60″ | 22°09′38.75″ |
| S6 | 113°32′57.06″ | 22°08′01.08″ |
| S7 | 113°32′55.44″ | 22°06′48.36″ |
| S8 | 113°33′37.26″ | 22°06′37.87″ |
| S9 | 113°34′30.60″ | 22°07′10.50″ |
| S10 | 113°35′51.65″ | 22°08′47.48″ |
| S11 | 113°34′11.02″ | 22°10′16.24″ |
| S12 | 113°33′36.60″ | 22°11′23.58″ |
| S13 | 113°35′40.00″ | 22°12′14.80″ |
| S14 | 113°36′36.80″ | 22°04′36.00″ |

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
