# Peer review of "Temporal and Spatial Distribution of Microplastics in a Coastal Region of the Pearl River Estuary, China"

_water, doi:10.3390/w13121618_

Round 1

Reviewer 1 Report

A good and interesting paper, which fits into the journal well.

They are some issues to be implemented:

Avoid lumping references. Instead, summarise the main contribution of each referenced paper in a separate sentence. Please carefully go through the entire manuscript.

Please improve the state of the art analysis to show the progress beyond state of the art clearly. The lack of proper justification creates the wrong impression that the authors are unaware of the recent developments. Please use relevant recent references by OTHER authors, recent meaning from 2020 and 2021.  This would provide the readers with a sense of continuity and help them place your paper in the context of what the journal has been publishing, very much strengthening your article's impact.

4 trillion USD - 

Remove "billion" as well as "trillion", as it is often commuted with "milliard" due to discrepancies between the UK and US English usage, use mathematical symbols instead.

 per cubic meter - better to use SI symbols

Table 2 - the references are from different years and the pollution has been developing (mostly growing) in different years - perhaps you should try to use the same time base.

Too many non-content words may indicate wordiness. Consider rewriting to avoid some of these words: the, for, of, into, via, as well as, their, with, are, also, which. For the text clarity, would you refrain from using additional words, mostly meaningless filler words, which can be omitted or some archaic words see, e.g. “respectively”, “thus”, “hence”, therefore”, “furthermore”, “thereby”, “basically,”, “meanwhile”,” wherein”, “herein”, “hitherto”, “Nonetheless”, “Perceivably”, “whereas”, etc.?

Have you considered changes and the impact caused by COVID-19 pandemic?

After dealing with the indicated suggestions the paper should be published.

Author Response

Date: May.22, 2021

Water

Dear Editor,

I am sending you herewith the revised manuscript of the paper entitled " Temporal and Spatial Distribution of Microplastics in a Coastal Region of the Pearl River Estuary, China (water-1240399) finished by my research group.

In the reversion, all the changes to the text have been highlighted in red. We have fully taken account of all the points made by the reviewers one by one. The details are listed at PDF file.

Sincerely

Chang Liu

Reviewer 2 Report

The authors present an interesting study on microplastics distribution in a coastal area of the Pearl River estuary. The study reviews the size, shape, and color of the MPs and their amount at different places. The article is worth publishing in ‘Water,’ however, several minor issues should be clear:

  • In the Introduction second paragraph from the end, it is written: “Globally, a total of 3.4 × 104 to 2.0 × 1010 MPs particles are released per day.” The range seems to be too large. Please explain or add a citation.
  • In Section 2.2: “sea water was used to rinse the net.” Why the authors used seawater that might contain MPs? Please add an explanation for how seawater is the water of choice.
  • Section 2.3: What is the sample volume? What are the purposes of the additives (ferrous sulfate solution hydrogen peroxide)?
  • Section 2.3: “this operation was repeated until the sample was clear” Clear of what?
  • Equation 1: Is the scattering of MPs on the membrane even? Doesn’t the filtering process change the regional distribution of the MPs on the membrane?
  • Below Figure 2: The authors should emphasize that only in S1—the river—shows higher MPs during the dry season. In my opinion, this proves that the source of the MPs is the river that its stream to the sea during the dry season is slower.
  • Below Table 1: Could it be that the MP sizes are too small to detect in the sea due to the weathering processes they went through? Meaning that in the sea, the MPs are too small to detect.
  • Figure 3 caption: What are the differences between a and b, c and d, e and f? The same in Figure 7.

Author Response

(The authors gave the same response as above.)

Reviewer 3 Report

Very interesting article on microplastics in aquatic environments. The objective of this study was to investigate the occurrence and distribution of MPs pollution in aquacultural water in the Pearl River Basin near the South China Sea. A well-formulated objective was achieved and correctly described. The paper reads with great interest.

Author Response

Thanks for your compliment.

Reviewer 4 Report

The manuscript water-1240399 entitled “Temporal and Spatial Distribution of Microplastics in a Coastal Region of the Pearl River Estuary, China” aimed to investigate the occurrence and distribution of microplastics pollution in aquacultural water in the Pearl River Basin near the South China Sea. The MS provides useful data to share with the scientific community. However, the M&M section need to be improved. Also, statistical analysis should be performed to compare microplastic abundance among sampling sites and seasons (dry and rain). Indeed, while I enjoyed the flow of the paper, I could not overcome the sense that there are some issues that could be addressed to improve the quality of the manuscript prior to publication in Water. 

I suggest to add line numbering during the revision process.

Specific comments

  • After “They have even been found in inaccessible regions of Antarctica [9] and the Arctic Ocean [10]” I suggest to add "high-mountain lakes" (reference: https://doi.org/10.1016/j.chemosphere.2020.129121)
  • I suggest to provide the geographical coordinates of the sampling sites (i.e., table as supplementary materials);
  • Section 2.2. How many replicates of water samples did you collect for each sampling site? Please, specify.
  • Section 2.5. I suggest to compare the MPs abundance among sampling sites. Please, use statistical tests. Also I suggest to compare the rain and dry seasons using statistical analysis.
  • I suggest to add a specific section on statistical analysis. Authors reported the results of principal component analysis but they did not mention it the section. Please, add a statement (why you chose to use it) in the M&M section.

Author Response

Dear Editor 

Thank you very much for your comments in a timely and professional manner. Those comments are all valuable and very helpful to revise and improve our paper, as well as the important guiding significance to our researches. We have considerably revised the manuscript based on these comments and responded the reviewers’ comments point-by-point.

Sincerely,

Dr. Chang Liu & Qijin Luo

South China Institute of Environmental Science, Ministry of Ecology and Environment

Guangzhou 510345, P. R. China.

E-mail address: liuchang@scies@org (C. Liu ), [email protected]

Round 2

Reviewer 2 Report

The authors took care of all my comments.

Author Response

Thank you very much for your comments in a timely and professional manner.

Reviewer 4 Report

The authors have addressed all my comments. The MS is now ready for publication in Water.